ecology/evolution/genetics

*Gadus morhua*, northeast Atlantic, selection, pantophysin, survival

**Author for correspondence:**
Håkon Otterå
e-mail: haakon.otteraa@hi.no

# The pantophysin gene and its relationship with survival in early life stages of Atlantic cod

Håkon Otterå[1], Torild Johansen[2], Arild Folkvord[1,3],
Geir Dahle[1], Marte Kristine Solvang Bingh[3],
Jon-Ivar Westgaard[2] and Kevin A. Glover[1,3]

[1]Institute of Marine Research, POB 1870, 5817 Bergen, Norway
[2]Institute of Marine Research, Tromsø Division, Framsenteret 9296 Tromsø, Norway
[3]Department of Biological Sciences, University of Bergen, Thormøhlensgt. 53, 5020 Bergen

 HO, 0000-0003-2450-4142; TJ, 0000-0002-9858-052X;
AF, 0000-0002-4763-0590; GD, 0000-0002-6070-8073;
J-IW, 0000-0002-9627-8124; KAG, 0000-0002-7541-1299

Genetic markers are widely used in fisheries management around the world. While the genetic structure and markers selected are usually based on samples from the wild, very few controlled experiments have been carried out to investigate possible differences in influence on traits between markers. Here we examine the bi-allelic gene pantophysin (Pan I), widely used in the management of Atlantic cod, in a series of *in vitro* crosses under a range of temperatures. It has been proposed that this gene, or another tightly linked gene, may be under strong divergent selection. Resolving this issue is essential in order to interpret results when using this gene marker for stock management. We found no evidence of departure from the expected $1:2:1$ Mendelian ratio for any of the three genotypes during the egg stage, while both the 6 and 12°C temperature regimes in tank experiments favoured the survival of the Pan I$^{AA}$ genotype. No difference in genotype survival was, however, found in a more natural mesocosm environment. Collectively, these results suggest that for the early life stages of Atlantic cod, and under the current experimental conditions, there is no strong consistent influence of Pan I genotype on survival. The results also emphasize the importance of varied experimental studies to verify the importance of environmental factors influencing genotype selection.

## 1. Introduction

Atlantic cod, *Gadus morhua* L., is a marine demersal fish that inhabits both sides of the Atlantic Ocean. In the Northeast Atlantic, it is found from the Bay of Biscay in the south to the ice edge in the Arctic ocean

in the north. This species has been and continues to be the target of major commercial fisheries in the northeast Atlantic [1,2], with annual landings of nearly 1 million tons (https://www.ices.dk). The population genetic structure of cod has been extensively studied during the last decades, and it is well established that Atlantic cod is divided into several genetic components on a large spatial scale [3–5]. There is also evidence for genetic population structure within national management units [6–8], and at a local scale (e.g. [9]).

In Norway, we have two major ecotypes of cod; (i) the highly migratory Northeast Arctic cod (NEAC), which has its feeding area mainly in the Barents sea but spawns along the Norwegian coast, and (ii) the more stationary Norwegian coastal cod (NCC) spawning along the coast and fjords [10]. The distinction between these groups has been acknowledged by fishermen far back in time. This distinction has also been important for the management of these stock components, as they both spawn in the same areas along the Norwegian coast during February to April, and give rise to a huge seasonal fishery before and during spawning [1,2]. The distinction between NEAC and NCC is important in this fishery because NEAC is numerically more abundant than NCC, and 'blind-harvest' without taking the separate stock components into consideration may lead to overexploitation of the less-abundant resource [11,12]. NEAC and NCC can be separated by morphological structure of the otoliths [13], a method that is still routinely used. However, genetic markers, including haemoglobin, allozymes, microsatellite and SNP markers have also been used to distinguish between NEAC and NCC (e.g. [1,8,14–20]). It is important to remember that the genetic assignment only gives the probability that a given specimen belongs to a population and is usually not diagnostic at the individual level [1,7].

Since the 1990s, the polymorphic locus pantophysin (Pan I, initially identified as synapthophysin, Syp I) has been widely used to characterize the genetic diversity of Atlantic cod [21,22]. Pantophysin is a membrane protein associated with cytoplasmic transport vesicles [23], although its function is not fully understood. Pan I is biallelic, even though there is some diversity within each of the two alleles [24]. In cod, and specifically between NEAC and NCC, this marker provides almost diagnostic discrimination between the two ecotypes. The NEAC is almost fixed for the Pan $I^{BB}$ genotype, and mainly the Pan $I^{AA}$ genotype is found in NCC [22,25,26]. This genetic marker has been used in real-time fisheries management to differentiate between the stock components (NEAC and NCC) in important spawning areas in Møre [1], and Lofoton [2] to protect spawning of the more vulnerable NCC.

A similar pattern in Pan I genotypes has also been observed from cod in Greenland, with a high proportion of the Pan $I^{BB}$ genotype in the offshore populations with assumed migrating cod, and mainly Pan $I^{AA}$ in the inshore samples (with assumed stationary cod) [27]. Also, in Icelandic waters, the Pan $I^{BB}$ variant is believed to undertake more migratory behaviour between feeding and spawning areas than the Pan $I^{AA}$ variant [28]. In the western part of the Atlantic, Georges Bank and in the coastal regions in Gulf of Main the situation is similar to the Barents Sea, with very high frequencies of the Pan I B allele [29]. Along the Norwegian coast, Pan I displays a gradient with decreasing frequency of the Pan $I^{AA}$ genotype towards the north and the Barents region, which could reflect the influence of interbreeding of the two ecotypes (NEAC and NCC) [6,7]. However, different allele frequencies at the Pan I locus are not observed between the cod captured in the North Sea versus the Norwegian coast [6,7], as the cod in the North Sea (as for the Faroe and Baltic cod) are almost fixed for Pan $I^{AA}$.

In recent years, SNP analysis has given us detailed knowledge of the cod genome. Analyses of the linkage map of more than 8000 SNPs revealed 23 major linkage groups for Atlantic cod, in which the Pan I locus is part of the chromosome or linkage group one (LG1) [8,18,20,30]. LG1 is the linkage group that has been the most studied in Atlantic cod, and important genetic differentiation between NEAC and NCC is located in here [19,20,31], while within three other linkage groups (LG2, LG7, LG12) SNP markers have been associated with temperature clines and other characteristics related to migration [8,18,19,32,33]. Two genome inversions (chromosome rearrangement of supergenes) located in the part of LG1 (including the Pan I locus), took place approximately 2 million years ago [19,20,33], and is observed in migrating cod across the North Atlantic [31]. Such inversions reduce recombination rates and therefore, in this case, help to maintain genetic differentiation between coastal cod and NEAC.

The fact that Pan I strongly differentiates between the two ecotypes in cod has been discussed for decades. Some assume the differences to be maintained through strong selection pressure on Pan I or related genes [33] either direct or indirect. Different selection responses to environmental conditions such as temperature, salinity and depth [26,34], and depth-related fishing pressure [35] have been suggested. Several studies have also reported a higher length at age of the Pan $I^{AA}$ compared to the Pan $I^{BB}$ [36–39], while also the opposite has been reported for 0-group juveniles [40]. Andersen *et al.* [41] analysed the phylogeny and various molecular aspects of Pan I and found no evidence for the

involvement of Pan I by itself in the different migratory activities of the cod populations. However, one of the genes linked to Pan I codes for rhodopsin, which is involved in photoreception [42] and has been suggested to be of potential importance for the vertical behaviour of cod [41,43]. Furthermore, among the 763 genes Kirubakaran *et al.* [20] modelled in the region containing the inversions in LG1 (and thereby linked to Pan I, see above), they also found several genes involved in key enzymes related to swim bladder function, haem synthesis and muscle organization which all may be of importance for the migratory behaviour. However, the differentiation between NEAC and NCC can also be explained by differences in breeding structure, as selection alone would be insufficient to cause the observed levels of genetic differentiation [44,45].

While it is generally accepted that Pan I somehow directly or indirectly is linked to functional differences between migratory and non-migratory cod ecotypes, including NEAC and NCC, the nature and magnitude of selection has only briefly been investigated through experimental work. Thus, there is still a need to elucidate variation in survival of the different genotypes of Pan I under different environmental conditions. Here, based upon a common garden experimental design, involving multiple families from the NCC ecotype, we investigated the relative survival and growth of offspring displaying three genotypes of Pan I: AA, AB and BB. We tested the Mendelian distribution at hatch, and early juvenile period, when mortality is high both under natural and experimental conditions, allowing for high selection pressure. Temperature is a key parameter for many processes and believed to be of importance for Pan I selection as discussed above. A rearing temperature comparison of 6 and 12°C was therefore used in controlled tank experiments, where 6°C represents a normal temperature during spawning both for NEAC and NCC [17,46], and 12°C represents the upper range of temperatures encountered for newly hatched Atlantic cod larvae. Experiments may be very sensitive to the experimental conditions and we therefore wanted to test the performance under more realistic environments. We therefore included two mesocosms in the set-up to mimic a more natural environment for the larvae, and thereby a selection pressure more similar to what coastal cod larvae may experience in nature. Finally, we contrasted the results from the different systems to see if the selection patterns were similar under fixed constant temperature conditions and seasonally varying temperature conditions during the early life stage of cod.

# 2. Material and methods

## 2.1. Overall study design

Two complementary common garden experiments using broodstock sampled at two different locations and years were performed by crossing adult cod of Pan I$^{AB}$ × Pan I$^{AB}$ genotypes (table 1). Offspring from both experiments were thereafter sampled and genotyped at hatch and at the juvenile stage and assigned to their families of origin. By the juvenile stage, a marine species like cod will have experienced significant mortality (e.g. [47]), allowing for testing of differential mortality between the genotypes, Pan I$^{AA}$, Pan I$^{AB}$ and Pan I$^{BB}$ compared to the expected 1 : 2 : 1 Mendelian distribution.

The parent fish used in both experiments were intended to belong to NCC, as opposed to NEAC, which can also be present in the same areas during spawning. The actual individuals used as broodstock for both experiment 1 and 2 (Exp. 1 and 2; table 1) were assigned to either NCC or NEAC by genotyping SNPs. The SNP analysis was done after the two experiments were terminated; therefore, we were not able to adjust the family set-up according to the assignment, but needed to exclude some of the crosses in the subsequent data analysis (see Results section and table 1). To do the SNP analysis, we included 100 NEAC from the Barents Sea and 100 NCC from the Lofoten region as reference samples for the assignment tests based on the program Geneclass [48]. The panel of markers used for genetic assignment comprised 28 neutral SNPs distributed across multiple linkage groups (LG2–LG16) to avoid the influence of LG1 (which includes the Pan gene) to the assignment result. Specific details about the analysis and assignment procedures are described in their entirety elsewhere [1].

Experiment 1 was performed in 2006 using tanks and employing two temperatures during the larval and juvenile rearing. Based on these results, it was decided to repeat the experiment with a more extensive set-up. In experiment 2, performed in 2014, both tank and mesocosm were applied. We incorporated as many family groups of larvae as possible in order to disentangle possible genetic effects not related to pantophysin. The tank experiment conditions in experiment 2 were the same as for experiment 1, including two temperatures. For the mesocosms natural temperature for Bergen

**Table 1.** Overview of the experiments. For each of the two experiments the family set-up (male × female) is indicated by code and genetical origin of the broodstock. Genetical origin was verified by SNP analysis. Further, the type of rearing units from yolk-sac larvae and onwards are indicated (tanks–6°C, tanks–12°C or mesocosm–ambient temperature), as well as the number of replicate rearing units. Date of sampling is also indicated. All fish were weighed at final sampling. * Few offspring in samples prevented further analysis of growth and selection in these families, and they were not included in any analyses or presentations. $ Not included in the analysis because the male was later assigned as NEAC.

| | broodstock | | | | rearing units | | | sampling | | |
| | female | | male | | no of replicates | | | | | |
| | | | | | tanks | | meso. | | | |
| family | code | origin | code | origin | 6°C | 12°C | amb. | yolk | met. | juv. |
|---|---|---|---|---|---|---|---|---|---|---|
| **Exp. 1** | | | | | | | | | | |
| 1A | F1 | NCC | M1 | NCC | — | 1-3-5 | — | 27 Apr | 2 Jun | 21 Aug |
| 1B | | | | | — | 1-3-3[a] | — | 27 Apr | 2 Jun | 21 Aug |
| 1C | | | | | 1 | 1 | — | 19 Apr | 2 Jun | 21 Aug |
| **Exp. 2** | | | | | | | | | | |
| 2* | F2 | NCC | M2 | NCC | 2 | 2 | 2 | 25 Mar | 15/22 May | — |
| 3* | | | M3 | —[b] | | | | | | |
| 4* | | | M4 | NCC[c] | | | | | | |
| 5$ | F3 | NCC | M5 | NEAC | | | | | | |
| 6 | | | M6 | NCC | | | | | | |
| 7 | | | M7 | NCC | | | | | | |

[a] one of the replicates were not genotyped.
[b] not assigned by SNP.
[c] male with genotype Pan I$^{AA}$.

region (60° N) was applied. A mesocosm is a much more complex and variable environment and resembles the natural environment more than smaller tank units [49].

In total, the two experiments allowed us to compare the relative survival between the Pan I genotypes from several families and at two temperatures in tank experiments as well as in a more natural mesocosm environment. The experimental design of the two experiments is detailed in table 1. All sampling and fish handling throughout the experiments were conducted by personnel trained according to the Norwegian legislation for animal experimentation.

## 2.2. Experiment 1

Broodstock for Exp. 1 were caught in Balsfjord, Northern Norway (69°18′ N,19°12′ E) in April 2005 and transported to the Institute of Marine Research (IMR) experimental facilities at Parisvatnet (60°37′ N, 4°48′ E). After genotyping 71 fish, individuals heterozygous for the Pan I gene were selected for the experiment in March 2006. These heterozygous broodstock were permitted to spawn naturally in tanks, and fertilized eggs were collected [50]. Five parent couples were Pan I heterozygotes; however, only one parent couple produced enough fertilized eggs in the time frame for the experiment. We selected three egg batches from this single family for the experiment, two batches spawned 30 March and one 3 April and hatched 19 and 27 April (batches were named family 1A, 1B and 1C, table 1).

Eggs from each batch were incubated in 180 l tanks supplied with *ca*. 1 l min$^{-1}$ seawater at 3°C. The larvae hatched after approximately three weeks and approximately 30 000 larvae from each batch were transported from Parisvatnet, to IMR's other marine research experimental facility at Austevoll, for further rearing in fish tanks. Larvae were enumerated from the density of larvae in the incubator; the water in the incubator was agitated to ensure a homogeneous larval distribution, and several (more than five) volumetric samples were taken and counted. The newly hatched larvae were thereafter transferred to 500 l tanks. Families 1A and 1B were kept in separate tanks at 12°C, while family batch 1C was split into two; one reared at 12°C and one at 6°C, enabling an explicit test of temperature effect on allele selection. During the summer, 1A and 1B were split into five and three tanks respectively, due to the growing biomass. The cold and warm groups from family 1C were held in their respective tanks until termination of the experiment in August.

For all three family batches, samples of 50–80 individuals for genotype determination were taken at hatch (April), shortly after metamorphosis (June) and at the juvenile stage at termination in August (table 1). Length and weight were also recorded at the final sample in August. The rearing protocol used in Exp. 1 was based on feeding cod larvae with cultured rotifers (*Brachionus plicatilis*) for the first weeks, followed by *Artemia* and thereafter weaned on formulated feed. The fish were fed in excess at all stages and reared under continuous (24 h) light.

## 2.3. Experiment 2

A new broodstock for Exp. 2 were caught off the coast of Finnmark, Northern Norway, October–November 2010 and thereafter transported to Parisvatnet. At the start of the experiment, March 2014, 26 of the broodstock were mature heterozygotes for Pan I and applicable for the experiment. In contrast with Exp. 1, the fish used as broodstock in Exp. 2 were manually stripped for eggs and milt on 11 March 2014 (table 1).

Atlantic cod held in captivity often become egg-bound [51,52], which means that they do not spawn and release the eggs, resulting in swollen abdomens and increased mortality. This was also a challenge in Exp. 2 and resulted in that only two egg batches (families) had the necessary amount and quality to be used in the experiment (table 1). Fertilization took place within 2 h of stripping. Each of the two egg batches were divided equally into three plastic jars, and each aliquot fertilized with 5 ml milt from one specific male (table 1). Milt was gently mixed with the eggs and seawater was thereafter added to activate the eggs. After 1 min, the three jars with fertilized eggs originating from one female were transferred to an egg incubator and incubated at approximately 5°C. The same was done with the egg batch from the second female. Incubation conditions were similar to those used in Exp. 1.

The number of larvae in each incubator was estimated at time of hatch, following the same procedure as for Exp. 1, approximately 15 days after fertilization. Based on the estimated larval density in each incubator, the desired volume was taken from each incubator and mixed together to make up the release group. This mixture was used to stock two mesocosms with approximately 100 000 larvae each, and four rearing tanks with approximately 40 000 larvae each.

The mesocosms (Meso A and B, $7 \times 7$ m wide, 3.5 m deep, approximately 175 m$^3$ volume) were suspended in a floating raft in Parisvatnet, a large (270 000 m$^3$ volume) enclosed natural seawater pond [53]. The mesocosms were made of woven polyamide, dark grey coloured and almost watertight. The cod larvae preyed on natural zooplankton in the bags, including rotifers, copepod nauplii and later copepodites, mainly from calanoid copepods. The large seawater pond in which the mesocoms were suspended acted as an additional, natural food supply for the rearing bags. The zooplankton was filtered from the pond and added to the rearing bags. From three weeks onwards, formulated feed (Gemma Micro followed by Gemma Wean and Gemma Diamond, www.skretting. com) was added to the rearing bags in addition to zooplankton. Temperatures in the mesocosms were ambient, increasing from 5°C at release to 13°C at harvest.

The larvae for the tank experiment were transported to Austevoll where they were reared under intensive rearing conditions, following approximately the same protocol as described for Exp. 1. Two of these tanks were held at a water temperature of 6°C and two at 12°C.

The two mesocosms were sampled on 15 May when the fish were approximately 0.4 g. Fish were caught by a casting net, resulting in a subsample of approximately 1500 individuals per mesocosm. At this time, the total number of juveniles was estimated to be approximately 2000 in Meso A and 5000– 10 000 in Meso B. The four tanks with intensively reared cod were emptied one week later and all survivors were frozen for later genotyping and weighing. Approximately 1000–2000 individuals had survived per tank until the time of sampling.

## 2.4. Genotyping and pedigree identification

DNA isolation of both broodstock and offspring resulting from both experiments was performed using the HotSHOT method [54]. In the first experiment, all the individuals were genotyped for Pan I through a standard PCR-RFLP method [22] with modified primers [55], and fragments separated on 2.5% MetaPhore gels, and scored manually. In experiment 2, all individuals were analysed for Pan I, and 10 microsatellites were used for assigning the individual fish to family. The Pan I primers are described in [56] and modified in [1]. The 10 microsatellite loci applied were Gmo2, Gmo3, Gmo8, Gmo19, Gmo34, Gmo35, Gmo37, Gmo132, Tch11, Tch13 [57–59]. The different fragments, including the Pan I fragments, were subsequently separated using an ABI 3130 XL sequencer (Applied Biosystems) and scored with the GeneMapper v. 5.0 software (Applied Biosystems). Pedigree was assigned using the exclusion-based method implemented in FAP [60].

## 2.5. Data analysis and statistics

Survival data (frequencies) were tested against a theoretical 1 : 2 : 1 Mendelian distribution using chi-square statistics [61]. To test if survival data from different families could be pooled within each rearing environment, Cochran–Mantel–Haenszel repeated test of independence was used [62]. Fish size data were analysed with mixed effects models using the nlme package in R [63] with tank as random factor. Higher-order non-significant terms were excluded from the model based on AIC criteria [64].

# 3. Results

## 3.1. Experiment 1

At the yolk-sac stage, three of the tanks displayed a Pan I genotype distribution that was in accordance with the expected 1-2-1 frequency (figure 1, left column). The fourth tank (family 1B, 12°C) was not in accordance with the expected $1 : 2 : 1$ distribution ($\chi^2$ test, $p < 0.01$, figure 1) and displayed a deficiency of individuals with the Pan I$^{BB}$ genotype. This was balanced with an increase both in the Pan I$^{AA}$ and Pan I$^{AB}$ genotype.

At the sampling point around metamorphosis, the three tanks that had experienced 12°C rearing temperature had similar genotype frequencies (figure 1, centre column). This was characterized with a deficiency of Pan I$^{BB}$ individuals and an excess of Pan I$^{AA}$ individuals, while the frequency of Pan I$^{AB}$ individuals was relatively unaffected compared to the frequency at yolk-sac stage. The tanks held at 12°C also had a very similar distribution to each other at the juvenile stage sampling point (figure 1, right column) and with a similar genotype distribution as they displayed at metamorphosis.

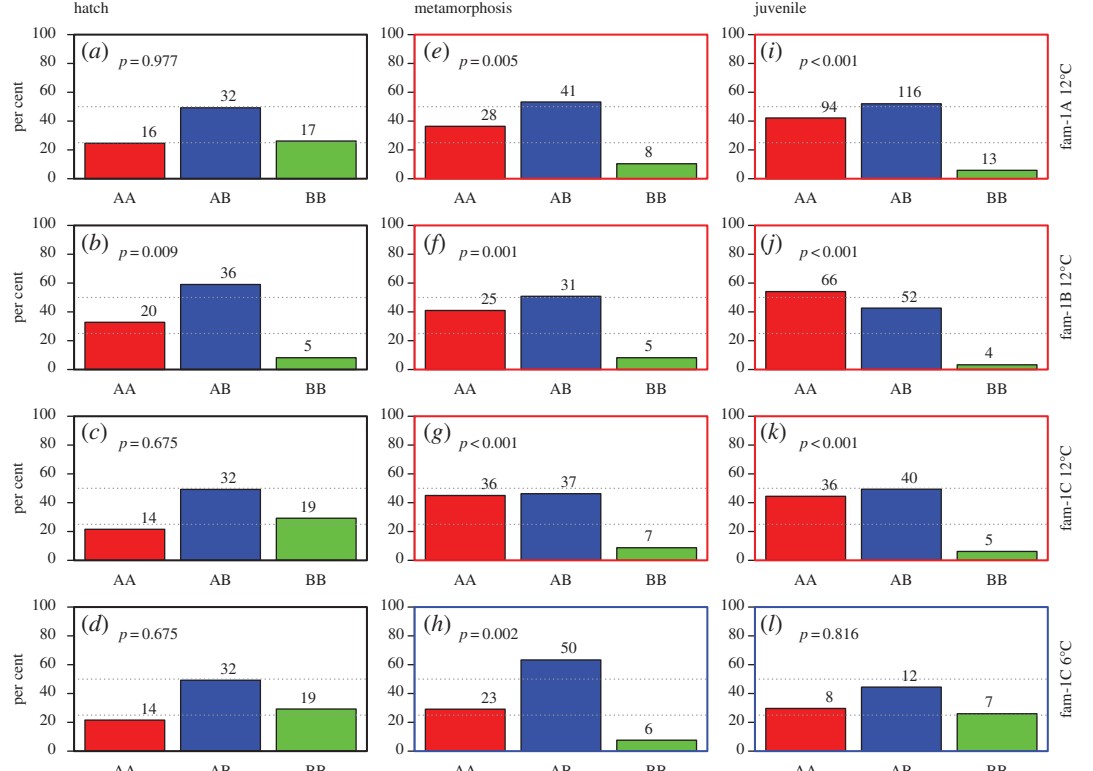

**Figure 1.** Overview of Pan I genotype distribution from Exp. 1. Each row of graphs represents offspring from one batch of eggs from the same family. The (a–d) graphs are the distribution sampled at the yolk-sac stage in the incubator, while the (e–h) and (i–l) graphs are from samples taken at metamorphosis and at juvenile stage respectively. Family 1C (annotated as family 1, batch C in table 1) was divided into two groups from yolk-sac stage and reared either at 12°C (red frames) or at 6°C (blue frames). The juvenile sample from families 1A and 1B are pooled numbers from five and three tanks, respectively. Genotype distribution is given as per cent, with actual counts written above each bar. p-values from the $\chi^2$ test, testing deviation from the expected $1:2:1$ ratio of genotypes.

The distribution pattern was slightly different for the single tank reared at 6°C from hatch and onwards. This one also had a deficiency of Pan $I^{BB}$ individuals at metamorphosis, but contrary to the three 12°C tanks, this was compensated by an increase in individuals with Pan $I^{AB}$, and not Pan $I^{AA}$. However, only 27 survived to the juvenile stage from this tank.

At the juvenile stage, the IC/cold group at 6°C weighed on average 1.5 g, as compared to 7.6 g for the three 12°C groups pooled. Genotype did not influence length nor log-weight in any of the groups ($p > 0.05$, mixed effect model).

## 3.2. Experiment 2

After genotyping, families 2 and 3 were only present in very low numbers both at hatch and metamorphosis (table 1). These were therefore excluded from subsequent analyses. Furthermore, family 4 lacked the Pan $I^{BB}$ genotype (table 1). This turned out to be caused by an error in the crossing design, where the sire was Pan $I^{AA}$ and not Pan $I^{AB}$ as intended. This family was therefore also excluded from the analyses. The SNP analyses that were performed after the experiments had been terminated also revealed that one of the males used in the crossings was NEAC and not NCC as intended. Crossing a NEAC, where the Pan gene is on an inversion with a NCC where the Pan gene is not on an inversion may reduce fertility and complicate the interpretation, and this family (family 5) was also excluded from the data analysis.

The Pan I genotype distribution at hatch did not deviate from the expected $1:2:1$ distribution (figure 2, families and replicates pooled). At metamorphosis, all three environments showed frequencies that deviated both from the expected Mendelian distribution, and from the observed distribution at hatch (distribution ($\chi^2$, $p < 0.01$, figure 2, families and replicates pooled). The cold and warm environments showed similar distributions at metamorphosis, with an over-representation of Pan $I^{AA}$ and fewer

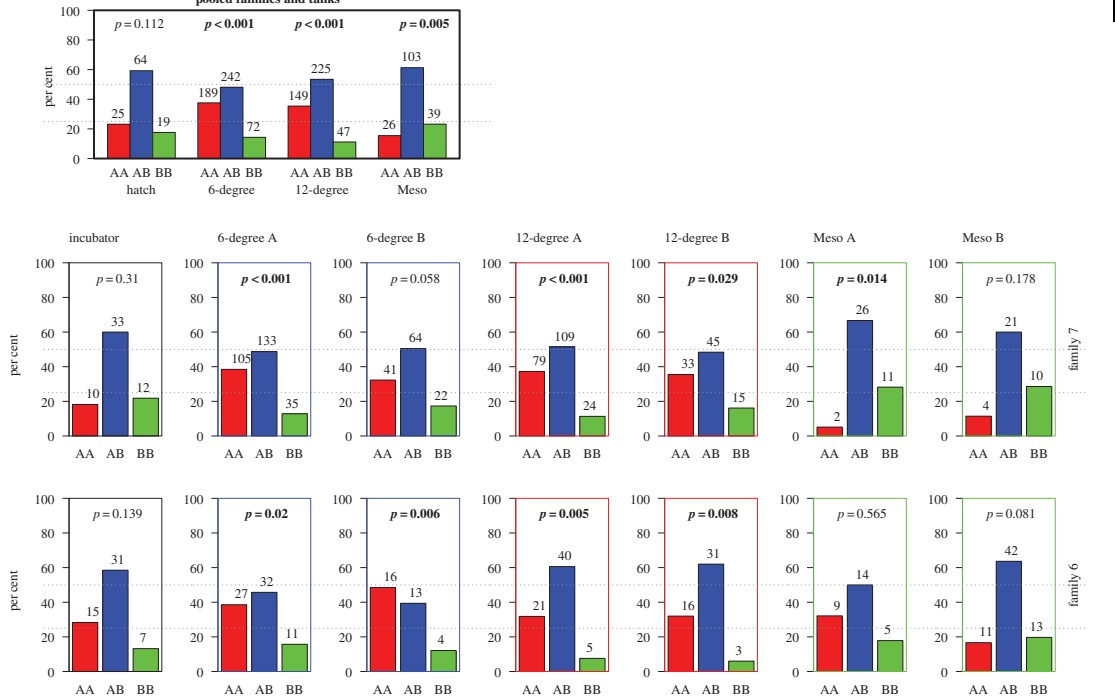

**Figure 2.** An overview of the Pan I genotype distribution from Exp. 2 are given at the top-left figure, where families and replicate tanks have been pooled. The distribution is shown at hatch (left) and at metamorphosis (6°C, 12°C, Meso). Number of individuals are shown above each bar, and $p$-values from the $\chi^2$ test, testing deviation from the expected 1 : 2 : 1 ratio of genotypes are given at the top. The smaller figures below show the genotype distribution and statistics for each rearing unit (column) and family (row) separately. We had one incubator at hatch, and two replicates for each environment at metamorphosis.

Pan I$^{BB}$, while Pan I$^{AB}$ was close to 50% as expected from a 1 : 2 : 1 Mendelian distribution (figure 2). For the mesocosm environment the situation was different, with more similar Pan I$^{AA}$ and Pan I$^{BB}$ frequencies and a slight overrepresentation of the heterozygote (figure 2).

The four combinations of family and cold temperature tanks measured at metamorphosis (figure 2) show some variation in genotype distribution between each other, but the overall distribution is similar. The same also applies to the warm temperature tanks and mesocosms. Furthermore, the interaction between Pan genotype and Family, taking Tank into consideration is only significant for the mesocosm environment (Cohran–Mantel–Haenzel repeated test of independence, $p = 0.03$), while the Pan I × Tank interaction, taking Family into consideration is not significant for any of the environments (Cohran–Mantel–Haenzel repeated test of independence, $p > 0.01$). Anyway, we suggest that pooling families/replicates within environment as shown in figure 2 is reasonable.

There was a considerable difference in weight at termination between fish from the different rearing environments ($p < 0.001$, mixed effects model, figure 3). The two intensive groups reared at 6 or 12°C had an average weight of 9 and 112 mg, respectively, at 58 days post hatch (both replicates pooled). The mesocosm groups sampled one week earlier had experienced a superior growth and had an average weight of 412 mg (both mesocosms pooled). A difference in weight between the three Pan I genotypes was evident, with AA being approximately 2% heavier than AB and 14% heavier than BB, mixed effects model ($p = 0.027$, figure 3). Family and group interactions were also significant, with family 4 being relatively heavier and family 6 being relatively lighter in the mesocosms compared to the relative family distribution in the laboratory groups ($p < 0.001$, Family × Group interaction).

## 4. Discussion

Despite the wealth of knowledge regarding the distribution of the three Pan I genotypes in cod, and its molecular structure and function, this is to our knowledge the first study where growth and survival of cod with the three Pan I genotypes have been compared in controlled experiments. We examined the frequency of the Pan I genotypes in one ecotype, NCC, against the expected Mendelian 1 : 2 : 1 distribution at hatch, at metamorphosis, and at the juvenile stages, while controlling for temperature

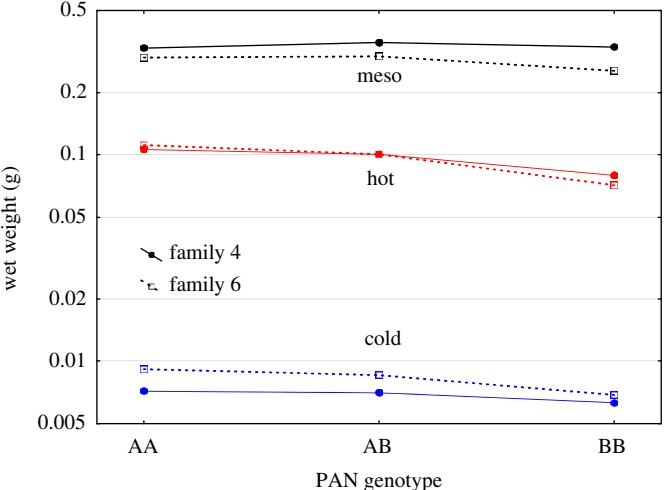

**Figure 3.** Modelled weight (g) of genotypes at termination in different groups and families from Exp. 2. Lines between points are included for visual clarity. The two replicates per environment have been pooled.

and food source/quality. We found some evidence of selection in favour of the Pan I$^{AA}$ genotype in the tank experiments conducted at 6 and 12°C. However, this was apparently contradicted by the results obtained in the mesocosm experiment, where selection favoured the Pan I$^{BB}$ and/or Pan I$^{AB}$ genotypes, even though the tank and mesocosm experiments had identical distribution of families at start. While at first these results may appear to conflict with each other, as they were conducted in different rearing environments, they illustrate that selection at this, or a strongly linked locus, is likely to be complicated and environment dependent.

In both experiments, the genotype distribution was almost perfect 1 : 2 : 1 at hatch. We found no clear evidence for selection for any of the Pan I genotypes at this stage, which includes possible selection during the fertilization and embryo stage. This is not unexpected, as mortality, at least in the hatchery is low during this stage, and with no predation or food limitation involved.

In marine species, including Atlantic cod, the period from hatch to metamorphosis is typically associated with high mortality, both under experimental conditions (e.g. [65]) and in the wild (e.g. [47]). Therefore, small differences in fitness between genotypes result in the significant potential to change their frequencies during such high mortality periods [66]. Here, we used two very different rearing environments during the larval period; one supposed to mimic a natural environment (mesocosms, Exp. 2 only) and the other, a more controlled tank environment at fixed temperature conditions (Exp. 1, Exp. 2).

The tank experiments gave similar relative survival at metamorphosis; a reduction in the Pan I$^{BB}$ variant, while the Pan I$^{AA}$ variant increased in frequency. This pattern was consistent for both tank experiments (Exp. 1, Exp. 2) and both temperatures. This suggests that within the scale of those tested, temperature alone cannot be a major driving force in selection on Pan I at this stage, since the Pan I$^{AA}$ genotype displayed highest survival under both temperature regimes. In Norwegian waters, NEAC and NCC overlap in several regions during the late winter and early spring spawning [1,2,67]. Temperature varies along the coast and is likely to play a role in determining the precise time [46] and potentially location of spawning. Here, a rearing temperature of 6°C was used as the low-temperature treatment, and this is in the upper part of what the wild cod (both NEAC and NCC) will experience during spawning in the northern parts of Norway [17]. It will be more representative of the lower end of the temperature range the cod will experience during the latter part of the larval and early parts of the juvenile stage, while the 12°C conditions will be at the high end of what they experience during these stages [68]. We can therefore not rule out that a lower rearing temperature during the larval period would have given a different result to that observed here. However, rearing cod at very low temperatures is much more challenging from a practical point of view and can give unrealistic results.

The results from the mesocosms (only Exp. 2) deviated from those from the tank experiments. In mesocosms, there was no evidence for a superior survival of the Pan I$^{AA}$ genotype. On the contrary, the Pan I$^{BB}$ variant had generally higher frequencies, but the differences were small. The mesocosm is a more diverse environment than an intensive tank system. The ambient space is much larger, the

food items may be more patchily distributed, competition between individual cod may be significant, and environmental factors such as light and temperature are varying. Thus, it is likely that also the selection factors operating are more complex and will be more difficult to disentangle.

As detailed in the introduction, the function of the Pan I gene is not fully understood, and its possible selection could be connected to the selection of other genes in the same linkage group [33]. The heterozygote variant Pan I$^{AB}$ was slightly overrepresented in the mesocosms, which was not observed in the tank experiments. An overdominance of Pan I heterozygotes in nature was reported by Karlsson & Mork [69] in the Trondheimsfjord when they studied differences between cohorts of NCC. However, they did not observe any allele frequency changes within the same cohort over time in adult cod. This could imply that selection that leads to heterozygote excess and differences in allele frequency occurs during the early life history when there is a high mortality. Excess of heterozygotes in nature does not necessarily imply a superior fitness, as it can also be caused, or supplemented by, crossing between genetically distinct populations with different Pan I allele frequencies, like NCC and NEAC. For instance, several genetic studies have reported an excess of Pan I heterozygotes in areas where NEAC and NCC mix during the annual spawning period [6,26,70].

We found differences in growth (weight at juvenile stage) between the genotypes in Exp. 2. However, this result must be treated with caution as growth differences when mortality is high may be the result of several mechanisms, including cannibalism. The results from Exp. 2, suggested highest weight of the Pan I$^{AA}$, followed by Pan I$^{AB}$, and Pan I$^{BB}$. This is in line with Case *et al.* [34] that did a similar mesocosm experiment and reported higher weight at 10 weeks after hatching of the Pan I$^{AB}$ compared to Pan I$^{BB}$ within each family, while the Pan I$^{AA}$ was not present in that experiment. Further, in most of the experimental work the coastal cod grow better than the NEAC under rearing conditions similar to what we used [71–74] which could relate to the Pan I genotype in NEAC.

An important part of the genetic difference between NCC and NEAC lays in the inversions of LG1, where the Pan I gene is located [8,20]. In turn, this makes it difficult to identify whether Pan I itself, or a neighbouring gene is causative of the biological differences between long-migrating NEAC and stationary NCC. The inversion effectively blocks or severely reduces the likelihood of recombination in this region of the genome. Thus, if NCC/Pan I$^{AA}$ cod mates with a NEAC/Pan I$^{BB}$ cod, their offspring will in turn produce gametes without recombination of genes within the inversion. Therefore, any potential linkage between Pan I and other possibly causative genes in this area will be retained and the offspring will be either an NCC or a NEAC and not something in between.

The extent of interbreeding between NCC and NEAC in nature is not clear [6,40,75], and a full discussion of this topic is beyond the scope of this paper and warrants further molecular research. Individuals from the two stocks may occupy the same spawning grounds, but different behaviour could restrict the breeding of significance [1,76]. In nature, Pan I heterozygotes are assumed to be present in both cod ecotypes [1,40], but whether some of them should be assumed hybrids is not clear. In laboratory experiments, they certainly can be crossed [74], but what ecotype such potential crossing between NEAC and NCC will be classified as, is more a philosophical question. Both migratory and stationary ecotypes were observed among Pan I heterozygotes in a large tagging study by Michalsen *et al.* [17], assigned to both NCC and NEAC.

Without including the mesocosms experiment in our study, the conclusion would have been that Pan I$^{AA}$ is superior to Pan I$^{BB}$, both regarding growth and survival during the larval and early juvenile period. Such results are consistent with the dominance of NCC in coastal areas in the south of Norway where the NCC is almost fixed for Pan I$^{AA}$. In these areas, the ambient temperature during the larval and juvenile period are similar to the experimental temperatures (6 and 12°C) we used in the tank experiments. Whether this pattern would have changed using even lower rearing temperature in the tanks remains an open question. Temperature in the mesocosm, increasing from 5 to 13°C during the larval period, is also more in line with what NCC in the south of Norway will meet during the same period in the wild. However, one should bear in mind that the NCC broodstock used in both our experiments were collected in the northernmost area in Norway, having low water temperatures and inhabited with both NCC and NEAC. It is very possible that the observed differences in mortality between the Pan I genotypes is caused by different growth rate among the genotypes. In other words, that we see a size-dependent mortality, where the fastest growing larvae and juveniles (in this case Pan I$^{AA}$) outcompetes the slightly slower growing Pan I$^{BB}$. Small differences in growth rate have previously been demonstrated to be important for relative survival among cod in a competitive environment [66]. These growth–survival connections are, as mentioned above, difficult to disentangle. The lack of superior survival of the Pan I$^{AA}$ variant in the mesocosms (with similar but varying temperature compared to the laboratory experiments), but

apparent higher growth also in this environment, suggest that the selection forces in a semi-natural environment such as a mesocosm are different from what the larvae experience in laboratory experiments and may be closer to what can be experienced in nature.

In summary, other environmental variables than the ambient temperature during the larval stages or selection mechanisms during later stages are likely to be responsible for the genetic variation we observe in Pan I in wild cod populations. Pan I is important for understanding the genetic difference observed between NEAC and NCC, and further experimental work on the interaction between environmental variables and the molecular genetic link to Pan I into the supergenes in LG 1 is strongly encouraged.

Ethics. All sampling and fish handling throughout the experiments were conducted by personnel trained according to the Norwegian legislation for animal experimentation.

Data accessibility. Supplementary data are available. Raw data showing genotype, length and weights from the two experiments are available from the Dryad Digital Repository: https://doi.org/10.5061/dryad.4xgxd255p [77].

Authors' contributions. All authors have contributed to the planning, experimental work/analysis and writing.

Competing interests. We declare we have no competing interests.

Funding. The study was financially supported by the Norwegian Ministry of Trade, Industry and Fisheries.

Acknowledgements. We appreciate the skilled technical support from technicians at IMR, Parisvatnet and IMR, Austevoll where the experiments were conducted. Further, the laboratory work by Anne Grete Eide Sørvik and Ole Ingar Paulsen is highly appreciated.

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
