## [Reviewer comments · Royal Society Open Science]

Review History

RSOS-191983.R0 (Original submission)

Review form: Reviewer 1

Is the manuscript scientifically sound in its present form?

Yes

Are the interpretations and conclusions justified by the results?

Yes

Is the language acceptable?

Yes

Do you have any ethical concerns with this paper?

No

Have you any concerns about statistical analyses in this paper?

Yes

Recommendation?

Major revision is needed (please make suggestions in comments)

Comments to the Author(s)

I have reviewed the manuscript entitled “The Pantophysin gene and its relationship with survival in Atlantic cod” by Håkon Otterå et al.

Overall, I think that it is important that these results be published. They represent a rare effort to understand functional adaptation due to differing PanI genotypes in Atlantic cod. I commend the work done by the authors, as common garden experiments are difficult to implement. However, as with many experiments, the results are not easy to explain or interpret, and I suggest improving clarity throughout the paper. The introduction would benefit from a clear objectives paragraph, which clearly describes the experimental design, what is the difference between a mesocosm and a tank, why temperatures were selected. State clearly why you performed both types of experiment (controlled and mesocosm). Justify the use of the 12° and 6°C temperatures. What are these aiming to mimic (You talk about this lines 374-376 but consider moving this to the introduction to justify your experimental design).

The results would benefit from more clarity, in how they are synthesized and expressed throughout the paper. It is important to remember that the results are limited to early life history stages; more changes could occur later in the development of the fish. Therefore, a more appropriate title might include: “with survival in early life stages of Atlantic cod”. It should also be emphasized in the abstract and conclusions that results are limited to early life stages. If PanI is important for migration, then it might be more relevant in life stages that are old enough to migrate.

Most readers are not as familiar with Atlantic cod as the authors of the paper. I have several questions that might guide revision from the standpoint of someone who is trying to understand NEAC and NCC in Atlantic cod:

1. You mention that Pantophysin is biallelic – does that mean that there are exactly two and only two versions of the gene? Or is it subject to minor mutations within each of the two main types, A and B?
2. Would an offspring with PanIBB whose parents are NCC and heterozygous for PanI be considered NEAC? Or mistakenly considered NEAC?
3. Broodstock were intended to be NCC – would this affect the experiment? What if you used NEAC heterozygous for Pan I? Would the results change?

Several other points include:

1. Why did you conduct the two experiments? Did you conduct Exp. 1 first and then decide it was not sufficient or was it part of a larger plan to do the experiment two ways? It can be difficult to separate Results and Methods in an experiment such as yours. Several sentences in the Material and Methods section are results: line 137-139, line 159, lines 174-175, lines 189-191, lines 201-202, lines 223-229, lines 231-235.
2. Can you describe more thoroughly what the fish experienced differently between Exp. 1 and Exp. 2? Did only food differ? It appears that temperature was controlled in both experiments. Anything else? Light? Water quality? You describe these in lines 389-392. Consider moving this up to the introduction.
3. Throughout your paper the NEAC and NCC are not presented as strongly distinct types as I would expect. They are not referred to as ecotypes, and in some places, you refer to “migratory and stationary cod” (e.g. line 103, line 120).
4. Do you implement a correction for multiple tests? For example, lines 278-279? Benjamini, Y. and Hochberg, Y., 1995. Controlling the false discovery rate: a practical and powerful approach to multiple testing. *Journal of the royal statistical society. Series B (Methodological)*, pp.289-300.
5. Were all offspring used in experiment 2 placed in a mesocosm? Lines 301-310 and Figure 2 are unclear whether the results in Figure 2 are entirely from Experiment 2 or both Exp. 1 and Exp 2. If Fig. 2 represents Exp. 1 and Exp. 2 then it should be clearly labeled. Remember that for publication in the journal selected, “a meaningful contribution to the scientific literature” is required. Be sure to promote your research as such. Currently the abstract does not indicate you are providing a meaningful contribution to scientific literature. Consider your work in a larger context and I think you will be able to emphasize its importance.

Other minor points:

Line 24: Consider: "...highly different abundances, which necessitates complex management strategies."

Line 28: Can you be more specific, such as: "In order to address the strength of selection,".

Line 33: consider "significantly" rather than "weakly". Your results were significant at the $p < 0.05$ level.

Line 35-36: True, but you might want to qualify this with also the life stages you observed. Also, after reading this sentence it appears that your research has no significance but that is not true.

Line 43: use "it is" rather than "it's" in formal scientific writing.

Line 44: consider "in the north".

Line 46: consider writing out million rather than "mill".

Line 48: it is.

Line 51: consider "...within countries (Dahle et al., 2018b), and at local scales..."

Line 52: consider removing "also".

Line 64: consider "to distinguish between"

Line 77: should be "has" rather than "have".

Lines 77-86: This is a great overview paragraph but it would benefit from being more specific regarding which ecotype you are talking about in each sentence. Is the sentence 79-81 about NEAC or NCC? Is the sentence from 81-84 about NCC? What ecotype are you referring about in the sentence from 84-86?

Line 94: Kirubakaran talks about several other characteristics besides temperature, considering prefacing this clause with "and other characteristics related to migration," or something like that.

Line 100: "will" not needed.

Line 109: This is something that was measured in your study. Did you collect lengths to confirm this? State this in your introduction when you describe goals of the experiment.

Line 242-246: Consider rewording – something like "In experiment 2, primers constructed for an ABI sequencer and Real-Time PCR were used..." and then describe.

Lines 266-267: This was also mentioned in lines 137-139.

Line 293-294: This belongs in the discussion. Why did they not survive? Could it be experimental design or something else?

Line 365: You state than in the lab experiment (Exp 1) there was a reduction in Pan IBB. But on line 368 you say this occurred in both experiments – do you mean Exp. 1 and Exp. 2, or at 6 and 12°C?

Line 387: You refer to Exp. 1 as "intensive conditions". Please be clear when referring to Exp.1 or 2.

Line 388: It may be confusing to some to refer to Pan IAA as "coastal". Be consistent – this is the common genotype in NCC. Consider using only "NCC" rather than "coastal".

Line 401: Do you mean allele frequency changes? Consider using the word "changes" rather than heterogeneity.

Line 406: Do you mean breeding between NCC and NEAC?

Lines 413-122: Include length here. Do you mean weight and length by "growth" or did you do any growth rate modeling and compare parameters?

Lines 424-432: Is there any evidence that NEAC and NCC mate or prefer not to mate? Does spawn timing coincide where they are found together?

Line 438: Which temperature – 6 or 12°C?

Line 440: Which difference in mortality? What higher growth rate? It is very important that you are clear on this point, as it helps the reader understand your results.

Line 445: Consider "may be" rather than "maybe".

Figure 2: Consider adding p-values to this plot, similar to Figure 1. Which sets represent metamorphosis?

Decision letter (RSOS-191983.R0)

Dear Dr Otterå,

The editors assigned to your paper ("The Pantophysin gene and its relationship with survival in Atlantic cod") have now received comments from reviewers. We would like you to revise your paper in accordance with the referee and Associate Editor suggestions which can be found below (not including confidential reports to the Editor). Please note this decision does not guarantee eventual acceptance.

Please submit a copy of your revised paper before 10-Jun-2020. Please note that the revision deadline will expire at 00.00am on this date. If we do not hear from you within this time then it will be assumed that the paper has been withdrawn. In exceptional circumstances, extensions may be possible if agreed with the Editorial Office in advance. We do not allow multiple rounds of revision so we urge you to make every effort to fully address all of the comments at this stage. If deemed necessary by the Editors, your manuscript will be sent back to one or more of the original reviewers for assessment. If the original reviewers are not available, we may invite new reviewers.

- Data accessibility

If you wish to submit your supporting data or code to Dryad (<http://datadryad.org/>), or modify your current submission to dryad, please use the following link:
<http://datadryad.org/submit?journalID=RSOS&manu=RSOS-191983>

- Competing interests

- Authors' contributions

- Acknowledgements

- Funding statement

on behalf of Professor Michael Bruford (Associate Editor) and Pete Smith (Subject Editor)
openscience@royalsociety.org

Reviewers' Comments to Author:

Reviewer: 1

Comments to the Author(s)

I have reviewed the manuscript entitled "The Pantophysin gene and its relationship with survival in Atlantic cod" by Håkon Otterå et al.

Overall, I think that it is important that these results be published. They represent a rare effort to understand functional adaptation due to differing PanI genotypes in Atlantic cod. I commend the work done by the authors, as common garden experiments are difficult to implement. However,

as with many experiments, the results are not easy to explain or interpret, and I suggest improving clarity throughout the paper. The introduction would benefit from a clear objectives paragraph, which clearly describes the experimental design, what is the difference between a mesocosm and a tank, why temperatures were selected. State clearly why you performed both types of experiment (controlled and mesocosm). Justify the use of the 12° and 6°C temperatures. What are these aiming to mimic (You talk about this lines 374-376 but consider moving this to the introduction to justify your experimental design).

The results would benefit from more clarity, in how they are synthesized and expressed throughout the paper. It is important to remember that the results are limited to early life history stages; more changes could occur later in the development of the fish. Therefore, a more appropriate title might include: "with survival in early life stages of Atlantic cod". It should also be emphasized in the abstract and conclusions that results are limited to early life stages. If PanI is important for migration, then it might be more relevant in life stages that are old enough to migrate.

Most readers are not as familiar with Atlantic cod as the authors of the paper. I have several questions that might guide revision from the standpoint of someone who is trying to understand NEAC and NCC in Atlantic cod:

1. You mention that Pantophysin is biallelic – does that mean that there are exactly two and only two versions of the gene? Or is it subject to minor mutations within each of the two main types, A and B?
2. Would an offspring with PanIBB whose parents are NCC and heterozygous for PanI be considered NEAC? Or mistakenly considered NEAC?
3. Broodstock were intended to be NCC – would this affect the experiment? What if you used NEAC heterozygous for Pan I? Would the results change?

Several other points include:

1. Why did you conduct the two experiments? Did you conduct Exp. 1 first and then decide it was not sufficient or was it part of a larger plan to do the experiment two ways? It can be difficult to separate Results and Methods in an experiment such as yours. Several sentences in the Material and Methods section are results: line 137-139, line 159, lines 174-175, lines 189-191, lines 201-202, lines 223-229, lines 231-235.
2. Can you describe more thoroughly what the fish experienced differently between Exp. 1 and Exp. 2? Did only food differ? It appears that temperature was controlled in both experiments. Anything else? Light? Water quality? You describe these in lines 389-392. Consider moving this up to the introduction.
3. Throughout your paper the NEAC and NCC are not presented as strongly distinct types as I would expect. They are not referred to as ecotypes, and in some places, you refer to "migratory and stationary cod" (e.g. line 103, line 120).
4. Do you implement a correction for multiple tests? For example, lines 278-279? Benjamini, Y. and Hochberg, Y., 1995. Controlling the false discovery rate: a practical and powerful approach to multiple testing. *Journal of the royal statistical society. Series B (Methodological)*, pp.289-300.
5. Were all offspring used in experiment 2 placed in a mesocosm? Lines 301-310 and Figure 2 are unclear whether the results in Figure 2 are entirely from Experiment 2 or both Exp. 1 and Exp 2. If Fig. 2 represents Exp. 1 and Exp. 2 then it should be clearly labeled.

Remember that for publication in the journal selected, "a meaningful contribution to the scientific literature" is required. Be sure to promote your research as such. Currently the abstract does not indicate you are providing a meaningful contribution to scientific literature. Consider your work in a larger context and I think you will be able to emphasize its importance.

Other minor points:

Line 24: Consider: "...highly different abundances, which necessitates complex management strategies."

Line 28: Can you be more specific, such as: "In order to address the strength of selection,".

Line 33: consider "significantly" rather than "weakly". Your results were significant at the $p < 0.05$ level.

Line 35-36: True, but you might want to qualify this with also the life stages you observed. Also, after reading this sentence it appears that your research has no significance but that is not true.

Line 43: use “it is” rather than “it’s” in formal scientific writing.

Line 44: consider “in the north”.

Line 46: consider writing out million rather than “mill”.

Line 48: it is.

Line 51: consider “...within countries (Dahle et al., 2018b), and at local scales...”

Line 52: consider removing “also”.

Line 64: consider “to distinguish between”

Line 77: should be “has” rather than “have”.

Lines 77-86: This is a great overview paragraph but it would benefit from being more specific regarding which ecotype you are talking about in each sentence. Is the sentence 79-81 about NEAC or NCC? Is the sentence from 81-84 about NCC? What ecotype are you referring about in the sentence from 84-86?

Line 94: Kirubakaran talks about several other characteristics besides temperature, considering prefacing this clause with “and other characteristics related to migration,” or something like that.

Line 100: “will” not needed.

Line 109: This is something that was measured in your study. Did you collect lengths to confirm this? State this in your introduction when you describe goals of the experiment.

Line 242-246: Consider rewording – something like “In experiment 2, primers constructed for an ABI sequencer and Real-Time PCR were used...” and then describe.

Lines 266-267: This was also mentioned in lines 137-139.

Line 293-294: This belongs in the discussion. Why did they not survive? Could it be experimental design or something else?

Line 365: You state than in the lab experiment (Exp 1) there was a reduction in Pan IBB. But on line 368 you say this occurred in both experiments – do you mean Exp. 1 and Exp. 2, or at 6 and 12°C?

Line 387: You refer to Exp. 1 as “intensive conditions”. Please be clear when referring to Exp.1 or 2.

Line 388: It may be confusing to some to refer to Pan IAA as “coastal”. Be consistent – this is the common genotype in NCC. Consider using only “NCC” rather than “coastal”.

Line 401: Do you mean allele frequency changes? Consider using the word “changes” rather than heterogeneity.

Line 406: Do you mean breeding between NCC and NEAC?

Lines 413-422: Include length here. Do you mean weight and length by “growth” or did you do any growth rate modeling and compare parameters?

Lines 424-432: Is there any evidence that NEAC and NCC mate or prefer not to mate? Does spawn timing coincide where they are found together?

Line 438: Which temperature – 6 or 12°C?

Line 440: Which difference in mortality? What higher growth rate? It is very important that you are clear on this point, as it helps the reader understand your results.

Line 445: Consider “may be” rather than “maybe”.

Figure 2: Consider adding p-values to this plot, similar to Figure 1. Which sets represent metamorphosis?

Author's Response to Decision Letter for (RSOS-191983.R0)

See Appendix A.

RSOS-191983.R1 (Revision)

Review form: Reviewer 1

Is the manuscript scientifically sound in its present form?

Yes

Are the interpretations and conclusions justified by the results?

Yes

Is the language acceptable?

Yes

Do you have any ethical concerns with this paper?

No

Have you any concerns about statistical analyses in this paper?

No

Recommendation?

Accept with minor revision (please list in comments)

Comments to the Author(s)

I have read the second version of the manuscript entitled "The Pantophysin gene and its relationship with survival in early life stages of Atlantic cod" by Otterå et al. The authors were very responsive to comments, and the current version is much improved. I have a few remaining concerns, but I do think this manuscript is on its way to a final version.

One primary concern is whether it is appropriate to include Family 5, as it is the offspring of an NEAC male. It may not be appropriate to include this family because other parts of the genome may have affected the results, even though the CMH test was not significant. This relates to a question I have regarding the extent that parents were analyzed for ecotype. Did the authors identify the inversion type for LG1, 2, 7 and 12 for each parent with SNP analysis? If so, did the NCC parents all have the "NCC type" for LG1, 2, 7 and 12? This is relevant to the question of how much other regions of the genome impacted the results.

In the discussion, paragraphs 2 and 3 discuss similarities and differences between Exp. 1 and 2. This is a very important part of the results because these experiments were conducted with almost identical experimental design. Any differences in results seem that they should come entirely from differences among families. Would a family effect or other differences in the genome possibly be an explanation for differences among experiments? Overall, I am a bit concerned about differences among families – are there other differences that are not being accounted for by simply looking at the pantophysin type?

I encourage the authors to specifically address the effect that temperature may have in the mesocosm experiments. There is some discussion of this, but was temperature monitored? Temperature and food differed between the tanks and it is important to disentangle any differences.

One last thing – is there any significance to such low spawning/breeding success? Is it possible that there is some genetic incompatibility among Pan1 heterozygotes? Models indicate that there should be very low spawning success among inversion heterozygotes. See reference below (Hoffmann and Rieseberg 2008).

Here is a list of specific comments:

Line 27: Consider "genetic" rather than "genetical". "Genetical" is a word, but less common than "genetic".

Line 29: Consider more specific wording – rather than “performance between markers” how about “influence on traits” “influence on selection”.

Line 29: Consider “Here we examine the bi-allelic gen, pantyphysin (Pan I) widely used in the management of Atlantic cod in a series of in vitro crosses under a range of temperatures. ”...

Line 31: Is “However” necessary?

Line 40: Consider moving “However” to the first word of the sentence.

Line 61: Consider “genetic population structure”.

Line 75: Here is another reference to support this: Spies, I. and Punt, A.E., 2015. The utility of genetics in marine fisheries management: a simulation study based on Pacific cod off Alaska. *Canadian Journal of Fisheries and Aquatic Sciences*, 72(9), pp.1415-1432.

Line 113: “are”?

Line 119: 1,310.

Line 127: Check journal convention for abbreviation of “million”, or just write it out.

Line 127: Kirubakaran (2016) states that LG1 contains a double inversion. Double inversions prevent double crossovers, almost completely suppressing homologous recombination in individuals heterozygous for the inversion.

Line 137: See this direct quote from Kirubakaran (2016): “The extreme divergence between NEAC and NCC at the Pan I locus has later been explained by differences in breeding structure, as selection alone would be insufficient to cause the observed levels of genetic differentiation (Fevolden & Sarvas 2001; Sarvas & Fevolden 2005; Westgaard & Fevolden 2007).”

Line 156: Since Pan IAA and Pan1BB types are on different inversion types, there is no recombination among heterozygotes – so mating among two heterozygotes may result in lower fertility.

See this from Hoffmann, A.A. and Rieseberg, L.H., 2008. Revisiting the impact of inversions in evolution: from population genetic markers to drivers of adaptive shifts and speciation?. *Annual review of ecology, evolution, and systematics*, 39, pp.21-42.

"Apart from TEs (transposable elements), another reason why the incidence of polymorphisms in large inversions might differ among species relates to the meiotic cost of infertility in inversion heterozygotes. When single crossovers occur in these heterozygotes, the recombinant gametes are unbalanced. However, in *Drosophila* females, three of the four gametes become polar cells and only one forms the egg; the recombinant gamete is less likely to become the egg because it migrates more slowly, reducing any cost associated with recombinant gametes. This factor coupled with the absence of recombination in males might explain the persistence of inversion polymorphisms in *Drosophila melanogaster* and other species, although it does not explain the high incidence of inversion polymorphisms in species where there is some male recombination (Krimbas & Powell 1992)."

Line 157: How about “survival” rather than “performance”?

Line 171: “Therefore”.

Line 196: Consider to “disentangle possible genetic effects not related to Pantyophysin”.

Line 196: I understand it was your intention to include several family groups, but forces beyond your control limited the number you could include. Consider rather than “Several” here, “We incorporated as many family groups as possible...”

Line 198: No period need after “Experiment 1”.

Line 197-198: State that you are referring to Experiment 2.

Line 202: “PAN” is a different abbreviation from elsewhere in the manuscript.

Line 205: “were”.

Line 210: “Marine”

Line 213: “eggs”

Line 214-215: This is interesting – because theoretically heterozygote crosses will have reduced fertility. Maybe the one family that resulted was not inversion heterozygotes for all 4 NCC/NEAC inversions.

Line 228: This “explicit test” is great – be sure to follow through in the discussion.

Line 240: Does continuous light mean light at night too?

Line 245-247: How were the broodstock identified for Exp2? Were they from the same 71 fish screen in Exp 1? Were the females and males also heterozygotes for Pan1?

Line 266: Write out approximately.

Line 277: Should read “filtered”.

Line 280: “Temperatures”

Line 313: add a space between “The” and “10”.

Lines 321-329: Consider moving this much earlier in the Methods section – as it applies to Exp 1 and 2. Once you did this, could you gather which inversion type each parent had on LG1, 2, 7 and 12?

Lines 332-338: What about the male that was actually not a heterozygote and the male that was NEAC? It makes sense to exclude those crosses or treat them separately.

Lines 349-352: For the discussion – could this be related to NCC parents – or NCC habitat/temperature?

Line 372: Shouldn't family 5 with a NEAC male be excluded?

Line 387: Showed.

Line 387-388: Please state what the pattern was – lower than expected Pan1BB?

Line 388: was.

Line 392: I am unsure whether it makes sense to pool a family with a NEAC male and families with NCC males. This implies that only the Pan1 genotype matters, as there are many genomic differences between NEAC and NCC. With a sample size of 1, this seems to be a potential

oversimplification of the results, although I know your Cochran Mantel Haenzel test did not show the results were different.

Line 417: consider adding... 'while controlling for temperature and food source/quality.'

Line 419: And these were bred from NCC adults -so maybe that makes sense.

Line 418-420: Did this come from different parents? Could there possibly have been a family effect?

Line 425-426: Here you could/should consider the effect of families.

Line 433: "results in".

Line 434-437: This sentence is off-topic with this paragraph - consider moving to next paragraph where you discuss proportion of genotypes at metamorphosis in Exp1 and 2. The intent of this paragraph seems to be to discuss the proportion of Pan1 genotypes at hatch in the tanks in Exp 1 and 2.

Lines 439-459: Why do you think the NCC genotype was favored in your experiment? Could it be because you had NCC parents (generally?). Or could it be due to the fact that you reared the fish in warmer temperature? Or water typically associated with NCC? It makes sense that the "NCC" type would be more successful in warmer temperatures, as it is a lower latitude ecotype - is this true? Would it be appropriate to discuss this point?

Line 447: Could this have anything to do with the fact that you started with NCC parents?

Line 473: There are 2 inversions adjacent to each other in LG1.

Line 479: Overdominance.

Line 481: spelling - necessarily.

Line 483: Wahlund effect. But in this case of a controlled experiment, it does not explain excess of heterozygotes.

Line 504-515: True. Did you control for this at all with your analysis of the parents? Could you tell which inversion type each parent had for the 2 inversions on LG1 and inversions on LG2, 7, and 12?

Line 522: "both cod" rather than "cod both".

Line 526: 'were'.

Line 529: In this paragraph, consider also the effect of temperature in the mesocosms. Did you measure temperature? What could it have been over the time period relative to the tanks?

Lines 540-560: Good handling of results when you do not exactly understand why.

Figures 1 and 2: Make these consistent; consider same y-axis label, and be consistent about whether you are labeling the bars. Add "yolk sac" and metamorphosis on Figure 2 somewhere, in addition to in the caption. In Figure 2, can you state that you only used data from families 5, 6, and 7 (if that is true?).

Table 1: For families 2, 3, and 4 from Exp. 2 - were any of these genotyped and included in Figure 2? Did you do anything with families 2, 3, and 4? Are there any results from these families?

Figure 3: Your legend is incomplete- what do the square, circle, and triangle represent?

Supplemental Figures: Please label Supplemental Figure 1 and Supplemental Figure 2 clearly in your working draft. I did not see captions for these in the .pdf provided. If there is room in the manuscript, consider moving Supplemental Figure 1 into the main body of the paper.

Decision letter (RSOS-191983.R1)

Dear Dr Otterå

On behalf of the Editors, we are pleased to inform you that your Manuscript RSOS-191983.R1 "The Pantophysin gene and its relationship with survival in early life stages of Atlantic cod" has been accepted for publication in Royal Society Open Science subject to minor revision in accordance with the referees' reports. Please find the referees' comments along with any feedback from the Editors below my signature.

Please submit your revised manuscript and required files (see below) no later than 7 days from today's (ie 11-Aug-2020) date. Note: the ScholarOne system will 'lock' if submission of the revision is attempted 7 or more days after the deadline. If you do not think you will be able to meet this deadline please contact the editorial office immediately.

on behalf of Professor Michael Bruford (Associate Editor) and Pete Smith (Subject Editor)
openscience@royalsociety.org

Reviewer comments to Author:

Reviewer: 1

Comments to the Author(s)

I have read the second version of the manuscript entitled “The Pantophysin gene and its relationship with survival in early life stages of Atlantic cod” by Otterå et al. The authors were very responsive to comments, and the current version is much improved. I have a few remaining concerns, but I do think this manuscript is on its way to a final version.

One primary concern is whether it is appropriate to include Family 5, as it is the offspring of an NEAC male. It may not be appropriate to include this family because other parts of the genome may have affected the results, even though the CMH test was not significant. This relates to a question I have regarding the extent that parents were analyzed for ecotype. Did the authors identify the inversion type for LG1, 2, 7 and 12 for each parent with SNP analysis? If so, did the NCC parents all have the “NCC type” for LG1, 2, 7 and 12? This is relevant to the question of how much other regions of the genome impacted the results.

In the discussion, paragraphs 2 and 3 discuss similarities and differences between Exp. 1 and 2. This is a very important part of the results because these experiments were conducted with almost identical experimental design. Any differences in results seem that they should come entirely from differences among families. Would a family effect or other differences in the genome possibly be an explanation for differences among experiments? Overall, I am a bit concerned about differences among families – are there other differences that are not being accounted for by simply looking at the pantophysin type?

I encourage the authors to specifically address the effect that temperature may have in the mesocosm experiments. There is some discussion of this, but was temperature monitored? Temperature and food differed between the tanks and it is important to disentangle any differences.

One last thing – is there any significance to such low spawning/breeding success? Is it possible that there is some genetic incompatibility among Pan1 heterozygotes? Models indicate that there should be very low spawning success among inversion heterozygotes. See reference below (Hoffmann and Rieseberg 2008).

Here is a list of specific comments:

Line 27: Consider “genetic” rather than “genetical”. “Genetical” is a word, but less common than “genetic”.

Line 29: Consider more specific wording – rather than “performance between markers” how about “influence on traits” “influence on selection”.

Line 29: Consider “Here we examine the bi-allelic gen, pantophysin (Pan I) widely used in the management of Atlantic cod in a series of in vitro crosses under a range of temperatures. ”...

Line 31: Is “However” necessary?

Line 40: Consider moving “However” to the first word of the sentence.

Line 61: Consider “genetic population structure”.

Line 75: Here is another reference to support this: Spies, I. and Punt, A.E., 2015. The utility of genetics in marine fisheries management: a simulation study based on Pacific cod off Alaska. Canadian Journal of Fisheries and Aquatic Sciences, 72(9), pp.1415-1432.

Line 113: “are”?

Line 119: 1,310.

Line 127: Check journal convention for abbreviation of “million”, or just write it out.

Line 127: Kirubakaran (2016) states that LG1 contains a double inversion. Double inversions prevent double crossovers, almost completely suppressing homologous recombination in individuals heterozygous for the inversion.

Line 137: See this direct quote from Kirubakaran (2016): “The extreme divergence between NEAC and NCC at the Pan I locus has later been explained by differences in breeding structure, as selection alone would be insufficient to cause the observed levels of genetic differentiation (Fevolden & Sarvas 2001; Sarvas & Fevolden 2005; Westgaard & Fevolden 2007).”

Line 156: Since Pan IAA and Pan1BB types are on different inversion types, there is no recombination among heterozygotes – so mating among two heterozygotes may result in lower fertility.

See this from Hoffmann, A.A. and Rieseberg, L.H., 2008. Revisiting the impact of inversions in evolution: from population genetic markers to drivers of adaptive shifts and speciation?. *Annual review of ecology, evolution, and systematics*, 39, pp.21-42.

"Apart from TEs (transposable elements), another reason why the incidence of polymorphisms in large inversions might differ among species relates to the meiotic cost of infertility in inversion heterozygotes. When single crossovers occur in these heterozygotes, the recombinant gametes are unbalanced. However, in *Drosophila* females, three of the four gametes become polar cells and only one forms the egg; the recombinant gamete is less likely to become the egg because it migrates more slowly, reducing any cost associated with recombinant gametes. This factor coupled with the absence of recombination in males might explain the persistence of inversion polymorphisms in *Drosophila melanogaster* and other species, although it does not explain the high incidence of inversion polymorphisms in species where there is some male recombination (Krimbas & Powell 1992)."

Line 157: How about “survival” rather than “performance”?

Line 171: “Therefore”.

Line 196: Consider to “disentangle possible genetic effects not related to Pantyophysin”.

Line 196: I understand it was your intention to include several family groups, but forces beyond your control limited the number you could include. Consider rather than “Several” here, “We incorporated as many family groups as possible...”

Line 198: No period need after “Experiment 1”.

Line 197-198: State that you are referring to Experiment 2.

Line 202: “PAN” is a different abbreviation from elsewhere in the manuscript.

Line 205: “were”.

Line 210: “Marine”

Line 213: “eggs”

Line 214-215: This is interesting – because theoretically heterozygote crosses will have reduced fertility. Maybe the one family that resulted was not inversion heterozygotes for all 4 NCC/NEAC inversions.

Line 228: This “explicit test” is great – be sure to follow through in the discussion.

Line 240: Does continuous light mean light at night too?

Line 245-247: How were the broodstock identified for Exp2? Were they from the same 71 fish screen in Exp 1? Were the females and males also heterozygotes for Pan1?

Line 266: Write out approximately.

Line 277: Should read “filtered”.

Line 280: “Temperatures”

Line 313: add a space between “The” and “10”.

Lines 321-329: Consider moving this much earlier in the Methods section – as it applies to Exp 1 and 2. Once you did this, could you gather which inversion type each parent had on LG1, 2, 7 and 12?

Lines 332-338: What about the male that was actually not a heterozygote and the male that was NEAC? It makes sense to exclude those crosses or treat them separately.

Lines 349-352: For the discussion – could this be related to NCC parents – or NCC habitat/temperature?

Line 372: Shouldn't family 5 with a NEAC male be excluded?

Line 387: Showed.

Line 387-388: Please state what the pattern was – lower than expected Pan1BB?

Line 388: was.

Line 392: I am unsure whether it makes sense to pool a family with a NEAC male and families with NCC males. This implies that only the Pan1 genotype matters, as there are many genomic differences between NEAC and NCC. With a sample size of 1, this seems to be a potential oversimplification of the results, although I know your Cochran Mantel Haenzel test did not show the results were different.

Line 417: consider adding...‘while controlling for temperature and food source/quality.’

Line 419: And these were bred from NCC adults –so maybe that makes sense.

Line 418-420: Did this come from different parents? Could there possibly have been a family effect?

Line 425-426: Here you could/should consider the effect of families.

Line 433: “results in”.

Line 434-437: This sentence is off-topic with this paragraph – consider moving to next paragraph where you discuss proportion of genotypes at metamorphosis in Exp1 and 2. The intent of this paragraph seems to be to discuss the proportion of Pan1 genotypes at hatch in the tanks in Exp 1 and 2.

Lines 439-459: Why do you think the NCC genotype was favored in your experiment? Could it be because you had NCC parents (generally?). Or could it be due to the fact that you reared the fish in warmer temperature? Or water typically associated with NCC? It makes sense that the “NCC” type would be more successful in warmer temperatures, as it is a lower latitude ecotype – is this true? Would it be appropriate to discuss this point?

Line 447: Could this have anything to do with the fact that you started with NCC parents?

Line 473: There are 2 inversions adjacent to each other in LG1.

Line 479: Overdominance.

Line 481: spelling – necessarily.

Line 483: Wahlund effect. But in this case of a controlled experiment, it does not explain excess of heterozygotes.

Line 504-515: True. Did you control for this at all with your analysis of the parents? Could you tell which inversion type each parent had for the 2 inversions on LG1 and inversions on LG2, 7, and 12?

Line 522: “both cod” rather than “cod both”.

Line 526: ‘were’.

Line 529: In this paragraph, consider also the effect of temperature in the mesocosms. Did you measure temperature? What could it have been over the time period relative to the tanks?

Lines 540-560: Good handling of results when you do not exactly understand why.

Figures 1 and 2: Make these consistent; consider same y-axis label, and be consistent about whether you are labeling the bars. Add “yolk sac” and metamorphosis on Figure 2 somewhere, in addition to in the caption. In Figure 2, can you state that you only used data from families 5, 6, and 7 (if that is true?).

Table 1: For families 2, 3, and 4 from Exp. 2 – were any of these genotyped and included in Figure 2? Did you do anything with families 2, 3, and 4? Are there any results from these families?

Figure 3: Your legend is incomplete- what do the square, circle, and triangle represent?

Supplemental Figures: Please label Supplemental Figure 1 and Supplemental Figure 2 clearly in your working draft. I did not see captions for these in the .pdf provided. If there is room in the manuscript, consider moving Supplemental Figure 1 into the main body of the paper.

===PREPARING YOUR MANUSCRIPT===

- one version identifying all the changes that have been made (for instance, in coloured highlight, in bold text, or tracked changes);
- a 'clean' version of the new manuscript that incorporates the changes made, but does not highlight them. This version will be used for typesetting.

===PREPARING YOUR REVISION IN SCHOLARONE===

-- Ensure that your data access statement meets the requirements at <https://royalsociety.org/journals/authors/author-guidelines/#data>. You should ensure that you cite the dataset in your reference list. If you have deposited data etc in the Dryad repository, please only include the 'For publication' link at this stage. You should remove the 'For review' link.

-- If you have uploaded ESM files, please ensure you follow the guidance at <https://royalsociety.org/journals/authors/author-guidelines/#supplementary-material> to include a suitable title and informative caption. An example of appropriate titling and captioning may be found at https://figshare.com/articles/Table_S2_from_Is_there_a_trade-off_between_peak_performance_and_performance_breadth_across_temperatures_for_aerobic_sc_ope_in_teleost_fishes_/3843624.

Author's Response to Decision Letter for (RSOS-191983.R1)

See Appendix B.

RSOS-191983.R2 (Revision)

Review form: Reviewer 1

Is the manuscript scientifically sound in its present form?

Yes

Are the interpretations and conclusions justified by the results?

Yes

Is the language acceptable?

Yes

Do you have any ethical concerns with this paper?

No

Have you any concerns about statistical analyses in this paper?

No

Recommendation?

Accept as is

Comments to the Author(s)

Thank you for responding to all my comments. I feel this manuscript is much improved and will make a valuable and interesting contribution to the literature.

Decision letter (RSOS-191983.R2)

Dear Dr Otterå,

It is a pleasure to accept your manuscript entitled "The Pantophysin gene and its relationship with survival in early life stages of Atlantic cod" in its current form for publication in Royal Society Open Science. The comments of the reviewer(s) who reviewed your manuscript are included at the foot of this letter.

Kind regards,
Lianne Parkhouse
Editorial Coordinator
Royal Society Open Science
openscience@royalsociety.org

on behalf of the Associate Editor, and Professor Pete Smith (Subject Editor)
openscience@royalsociety.org

Associate Editor Comments to Author:

Thank you for seriously engaging with the reviewer's comments.

Reviewer comments to Author:
Reviewer: 1
Comments to the Author(s)

Thank you for responding to all my comments. I feel this manuscript is much improved and will make a valuable and interesting contribution to the literature.

Appendix A

Dear editor

Thank you for revising the manuscript. We have found the referees comments and suggestions very useful and have made changes to the documents according to referee's suggestion. A detailed explanation of this is shown below. This is a paper describing a relatively complicated experimental setup, and we have through the revision tried to make the presentation easier to read and understand. We have also followed the other suggestions from the referee and hope the manuscript now has a broader scope and will be interesting for the journal's readers.

Sincerely

Håkon Otterå

on behalf of the authors

DETAILED COMMENTS TO REFFEREE REPORT

Original referee text in normal font

Our comments to each point in italic font

I have reviewed the manuscript entitled "The Pantophysin gene and its relationship with survival in Atlantic cod" by Håkon Otterå et al.

Overall, I think that it is important that these results be published. They represent a rare effort to understand functional adaptation due to differing PanI genotypes in Atlantic cod. I commend the work done by the authors, as common garden experiments are difficult to implement. However, as with many experiments, the results are not easy to explain or interpret, and I suggest improving clarity throughout the paper.

We have made significant changes to the document to make it easier to understand, particularly Abstract, Introduction and MM. Be also aware that Table 1 is supposed to help understand the experimental setup.

The introduction would benefit from a clear objectives paragraph, which clearly describes the experimental design, what is the difference between a mesocosm and a

tank, why temperatures were selected. State clearly why you performed both types of experiment (controlled and mesocosm). Justify the use of the 12° and 6°C temperatures. What are these aiming to mimic (You talk about this lines 374-376 but consider moving this to the introduction to justify your experimental design).

The introduction has been changed, and objective of the experiments are now more clearly expressed.

The results would benefit from more clarity, in how they are synthesized and expressed throughout the paper. It is important to remember that the results are limited to early life history stages; more changes could occur later in the development of the fish.

Therefore, a more appropriate title might include: “with survival in early life stages of Atlantic cod”. It should also be emphasized in the abstract and conclusions that results are limited to early life stages. If PanI is important for migration, then it might be more relevant in life stages that are old enough to migrate.

“Early life stage” are now used in the title and at relevant places throughout the paper. The MM have been changed to make the experimental design easier to understand.

Most readers are not as familiar with Atlantic cod as the authors of the paper. I have several questions that might guide revision from the standpoint of someone who is trying to understand NEAC and NCC in Atlantic cod:

1. You mention that Pantophysin is biallelic – does that mean that there are exactly two and only two versions of the gene? Or is it subject to minor mutations within each of the two main types, A and B? *Changes made line 77.*
2. Would an offspring with PanIBB whose parents are NCC and heterozygous for PanI be considered NEAC? Or mistakenly considered NEAC? *This has now been more extensively covered in the discussion, and commented directly, L464x.*
3. Broodstock were intended to be NCC – would this affect the experiment? What if you used NEAC heterozygous for Pan I? Would the results change? *This is a big question that has not been directly answered in the paper, but we have written about connected genes, inversion and things are related to the question. We have tried to “isolate” the PAN I function and test performance of PAN per se, but to say much about what would happen with a different genetical structure (beside*

(PAN) is a question that is almost impossible to answer.

Several other points include:

1. Why did you conduct the two experiments? Did you conduct Exp. 1 first and then decide it was not sufficient or was it part of a larger plan to do the experiment two ways? It can be difficult to separate Results and Methods in an experiment such as yours. Several sentences in the Material and Methods section are results: line 137-139, line 159, lines 174-175, lines 189-191, lines 201-202, lines 223-229, lines 231-235. *This has now been clarified in the beginning of MM. Some of the text suggested as Results have been moved from MM to the Results section. However, it is difficult to present the setup without including some “results”, and in these cases we have kept it in MM. This is a type of result that is not an important part of the experimental design in this experiment.*
2. Can you describe more thoroughly what the fish experienced differently between Exp. 1 and Exp. 2? Did only food differ? It appears that temperature was controlled in both experiments. Anything else? Light? Water quality? You describe these in lines 389-392. Consider moving this up to the introduction. *Both Introduction and MM has been revised so that the experimental conditions should be easier to understand.*
3. Throughout your paper the NEAC and NCC are not presented as strongly distinct types as I would expect. They are not referred to as ecotypes, and in some places, you refer to “migratory and stationary cod” (e.g. line 103, line 120). *We have used “ecotypes” more consequent in the revision.*
4. Do you implement a correction for multiple tests? For example, lines 278-279? Benjamini, Y. and Hochberg, Y., 1995. Controlling the false discovery rate: a practical and powerful approach to multiple testing. *Journal of the royal statistical society. Series B (Methodological)*, pp.289-300. *No, such corrections were not used, and their use are debated. We had only very few (2-3) multiple comparisons within a test, and the correction would anyway have no consequences on the interpretation. All p-values are given in figures/suppl-figures so it will be possible for interested readers to check the results also with corrections.*
5. Were all offspring used in experiment 2 placed in a mesocosm? Lines 301-310 and Figure 2 are unclear whether the results in Figure 2 are entirely from Experiment 2 or both Exp. 1 and Exp 2. If Fig. 2 represents Exp. 1 and Exp. 2

then it should be clearly labeled.

This is probably due to a misunderstanding regarding differences between exp 1 and 2, which now hopefully are clarified by the changes mentioned above.

Figure 1 and 2 are clearly labeled as they are – and represent exp 1 and 2 respectively.

Remember that for publication in the journal selected, “a meaningful contribution to the scientific literature” is required. Be sure to promote your research as such. Currently the abstract does not indicate you are providing a meaningful contribution to scientific literature. Consider your work in a larger context and I think you will be able to emphasize its importance.

Abstract has been revised, and has now a wider perspective.

Other minor points:

Line 24: Consider: “...highly different abundances, which necessitates complex management strategies.” *Done*

Line 28: Can you be more specific, such as: “In order to address the strength of selection.” *Done*

Line 33: consider “significantly” rather than “weakly”. Your results were significant at the $p < 0.05$ level. *Done*

Line 35-36: True, but you might want to qualify this with also the life stages you observed. Also, after reading this sentence it appears that your research has no significance but that is not true. *Changes made.*

Line 43: use “it is” rather than “it’s” in formal scientific writing.

Done

Line 44: consider “in the north”. *Done*

Line 46: consider writing out million rather than “mill”.

Done

Line 48: it is. *Done*

Line 51: consider “...within countries (Dahle et al., 2018b), and at local scales...”

Done

Line 52: consider removing “also”. *Done*

Line 64: consider “to distinguish between”

Done

Line 77: should be “has” rather than “have”.

Done

Lines 77-86: This is a great overview paragraph but it would benefit from being more specific regarding which ecotype you are talking about in each sentence. Is the sentence 79-81 about NEAC or NCC? Is the sentence from 81-84 about NCC? What ecotype are you referring about in the sentence from 84-86? *Changes made*

Line 94: Kirubakaran talks about several other characteristics besides temperature, considering prefacing this clause with “and other characteristics related to migration,” or something like that. *Changes made*

Line 100: “will” not needed. *Done*

Line 109: This is something that was measured in your study. Did you collect lengths to confirm this? State this in your introduction when you describe goals of the experiment. *Now stated explicitly in the introduction*

Line 242-246: Consider rewording – something like “In experiment 2, primers constructed for an ABI sequencer and Real-Time PCR were used...” and then describe. *Changes made*

Lines 266-267: This was also mentioned in lines 137-139. *This is an important part of the setup and was therefore mentioned in the beginning of MM help the reader understand the setup. It also needs to be mentioned in the statistics section.*

Line 293-294: This belongs in the discussion. Why did they not survive? Could it be experimental design or something else?

Has been rephrased, but not included as a specific point in the discussion. The challenges with rearing in tanks, particularly at low temperatures is however, mentioned in the discussion.

Line 365: You state than in the lab experiment (Exp 1) there was a reduction in Pan IBB. But on line 368 you say this occurred in both experiments – do you mean Exp. 1 and Exp. 2, or at 6 and 12°C? *Has been clarified.*

Line 387: You refer to Exp. 1 as “intensive conditions”. Please be clear when referring to Exp.1 or 2. *Has been clarified*

Line 388: It may be confusing to some to refer to Pan IAA as “coastal”. Be consistent – this is the common genotype in NCC. Consider using only “NCC” rather than “coastal”. *Done*

Line 401: Do you mean allele frequency changes? Consider using the word “changes” rather than heterogeneity. *Done*

Line 406: Do you mean breeding between NCC and NEAC? *Yes, changed*

Lines 413-122: Include length here. Do you mean weight and length by “growth” or did you do any growth rate modeling and compare parameters? *Clarified*

Lines 424-432: Is there any evidence that NEAC and NCC mate or prefer not to mate? Does spawn timing coincide where they are found together? *This is a big topic that we consider outside the scope of this paper. We have therefore only mentioned this very briefly in the revision.*

Line 438: Which temperature – 6 or 12°C? *Rephrased*

Line 440: Which difference in mortality? What higher growth rate? It is very important that you are clear on this point, as it helps the reader understand your results. *Have written more about this, so it should be clearer.*

Line 445: Consider “may be” rather than “maybe”. *Done*

Figure 2: Consider adding p-values to this plot, similar to Figure 1. Which sets represent metamorphosis? *Figure 2 refers to Suppl 1 for more details, and in Suppl 1 the individual (per family and replicate) p-values are shown. [It could be that the two supplementary figures have not been included in the material for the referee – even though they were uploaded?]. Have changed figure 2 caption to clarify what bars that represents metamorphosis.*

Appendix B

28 August 2020

Once more, thank you for a very thorough and useful review. We have now made a new version of the manuscript, incorporating the comments made by the reviewer.

The main change is that we have removed one of the families from the analysis and presentation, as suggested. The relevant statistics have been rerun accordingly, and Table and Figures also corrected (the removed family is still present in the data repository, and easily identified there). We have also “merged” one of the Supplementary figures into the Figures, as also suggested. The other Suppl. is less relevant with one less family and has been deleted. We have also included more information/discussion regarding temperature aspects and clarified some of the paragraphs to avoid potential misunderstandings.

A more detailed description of the changes made are written in *italic* under, as response to each point made by the reviewer.

Reviewer: 1

Comments to the Author(s)

I have read the second version of the manuscript entitled “The Pantophysin gene and its relationship with survival in early life stages of Atlantic cod” by Otterå et al. The authors were very responsive to comments, and the current version is much improved. I have a few remaining concerns, but I do think this manuscript is on its way to a final version.

One primary concern is whether it is appropriate to include Family 5, as it is the offspring of an NEAC male. It may not be appropriate to include this family because other parts of the genome may have affected the results, even though the CMH test was not significant. This relates to a question I have regarding the extent that parents were analyzed for ecotype. Did the authors identify the inversion type for LG1, 2, 7 and 12 for each parent with SNP analysis? If so, did the NCC parents all have the “NCC type” for LG1, 2, 7 and 12? This is relevant to the question of how much other regions of the genome impacted the results.

Family 5 has now been removed. Unfortunately, we do not have information about the different inversions for each parent, and this is anyway not a trivial analysis to do.

In the discussion, paragraphs 2 and 3 discuss similarities and differences between Exp. 1 and 2. This is a very important part of the results because these experiments were conducted with almost identical experimental design. Any differences in results seem that they should come entirely from differences among families. Would a family effect or other differences in the genome possibly be an explanation for differences among experiments? Overall, I am a bit concerned about differences among families – are there other differences that are not being accounted for by simply looking at the pantophysin type?

The two experiments gave quite similar results, while the main difference in survival was between the tank- and mesocosm environment (both part of exp 2 – and with identical family distribution at start). I think these things are more clearly described now, to avoid misunderstandings.

I encourage the authors to specifically address the effect that temperature may have in the mesocosm experiments. There is some discussion of this, but was temperature monitored? Temperature and food differed between the tanks and it is important to disentangle any differences.

Temperature is one of the major topics in the discussion, and some more information have been added.

'One last thing – is there any significance to such low spawning/breeding success? Is it possible that there is some genetic incompatibility among Pan1 heterozygotes? Models indicate that there should be very low spawning success among inversion heterozygotes. See reference below (Hoffmann and Rieseberg 2008).

The main reason for the low breeding success was the “egg-bound” problem which is very common in captive cod. Further, by excluding the NEAC male (Family 5) all parents were NCC and NEAC/NCC heterozygotes were avoided.

Here is a list of specific comments:

OK means that changes were made as suggested.

Line 27: Consider “genetic” rather than “genetical”. “Genetical” is a word, but less common than “genetic”. *OK*

Line 29: Consider more specific wording – rather than “performance between markers” how about “influence on traits” “influence on selection”. *OK*

Line 29: Consider “Here we examine the bi-allelic gen, pantyphysin (Pan I) widely used in the management of Atlantic cod in a series of in vitro crosses under a range of temperatures.” ...

OK

Line 31: Is “However” necessary?

OK

Line 40: Consider moving “However” to the first word of the sentence. *OK*

Line 61: Consider “genetic population structure”. *OK*

Line 75: Here is another reference to support this: Spies, I. and Punt, A.E., 2015. The utility of genetics in marine fisheries management: a simulation study based on Pacific cod off Alaska. Canadian Journal of Fisheries and Aquatic Sciences, 72(9), pp.1415-1432. *OK*

Line 113: “are” *OK*

Line 119: 1,310. *OK*

Line 127: Check journal convention for abbreviation of “million”, or just write it out. *OK*

Line 127: Kirubakaran (2016) states that LG1 contains a double inversion. Double inversions prevent double crossovers, almost completely suppressing homologous recombination in individuals heterozygous for the inversion.

Double inversion specified

Line 137: See this direct quote from Kirubakaran (2016): “The extreme divergence between NEAC and NCC at the Pan I locus has later been explained by differences in breeding structure, as selection alone would be insufficient to cause the observed levels of genetic differentiation (Fevolden & Sarvas 2001; Sarvas & Fevolden 2005; Westgaard & Fevolden 2007).”

Sentence regarding this added at the end of section

Line 156: Since Pan IAA and Pan1BB types are on different inversion types, there is no recombination among heterozygotes – so mating among two heterozygotes may result in lower fertility.

This is more clearly stated now

See this from Hoffmann, A.A. and Rieseberg, L.H., 2008. Revisiting the impact of inversions in evolution: from population genetic markers to drivers of adaptive shifts and speciation?. *Annual review of ecology, evolution, and systematics*, 39, pp.21-42.

"Apart from TEs (transposable elements), another reason why the incidence of polymorphisms in large inversions might differ among species relates to the meiotic cost of infertility in inversion heterozygotes. When single crossovers occur in these heterozygotes, the recombinant gametes are unbalanced. However, in *Drosophila* females, three of the four gametes become polar cells and only one forms the egg; the recombinant gamete is less likely to become the egg because it migrates more slowly, reducing any cost associated with recombinant gametes. This factor coupled with the absence of recombination in males might explain the persistence of inversion polymorphisms in *Drosophila melanogaster* and other species, although it does not explain the high incidence of inversion polymorphisms in species where there is some male recombination (Krimbas & Powell 1992)."

This “point” I think is incorporated in the revised text

Line 157: How about “survival” rather than “performance”?

OK

Line 171: “Therefore”.

OK

Line 196: Consider to “disentangle possible genetic effects not related to Pantyophysin”.

OK

Line 196: I understand it was your intention to include several family groups, but forces beyond your control limited the number you could include. Consider rather than “Several” here, “We incorporated as many family groups as possible...”

OK

Line 198: No period need after “Experiment 1”.

OK

Line 197-198: State that you are referring to Experiment 2.

OK

Line 202: “PAN” is a different abbreviation from elsewhere in the manuscript.

OK

Line 205: “were” . OK

Line 210: “Marine” OK

Line 213: “eggs” OK

Line 214-215: This is interesting – because theoretically heterozygote crosses will have reduced fertility. Maybe the one family that resulted was not inversion heterozygotes for all 4 NCC/NEAC inversions.

Maybe, but also in natural spawning (exp 1) the “egg-bound” problem and spawning behavior makes it difficult to achieve all crossings that you want in an experiment. This often make it difficult to setup a perfect experiment where such things can be tested.

Line 228: This “explicit test” is great – be sure to follow through in the discussion.

Have tried to do that

Line 240: Does continuous light mean light at night too? OK

Line 245-247: How were the broodstock identified for Exp2? Were they from the same 71 fish screen in Exp 1? Were the females and males also heterozygotes for Pan1?

Text rephrased

Line 266: Write out approximately. OK

Line 277: Should read “filtered”. OK

Line 280: “Temperatures” OK

Line 313: add a space between “The” and “10”. OK

Lines 321-329: Consider moving this much earlier in the Methods section – as it applies to Exp 1 and 2. Once you did this, could you gather which inversion type each parent had on LG1, 2, 7 and 12?

Text moved

Lines 332-338: What about the male that was actually not a heterozygote and the male that was NEAC? It makes sense to exclude those crosses or treat them separately.

Excluded

Lines 349-352: For the discussion – could this be related to NCC parents – or NCC habitat/temperature?

This is covered in the discussion

Line 372: Shouldn't family 5 with a NEAC male be excluded?

Excluded

Line 387: Showed.

OK

Line 387-388: Please state what the pattern was – lower than expected Pan1BB?

OK

Line 388: was.

OK

Line 392: I am unsure whether it makes sense to pool a family with a NEAC male and families with NCC males. This implies that only the Pan1 genotype matters, as there are many genomic differences between NEAC and NCC. With a sample size of 1, this seems to be a potential oversimplification of the results, although I know your Cochran Mantel Haenzel test did not show the results were different.

Family excluded

Line 417: consider adding...'while controlling for temperature and food source/quality.'

OK

Line 419: And these were bred from NCC adults –so maybe that makes sense.

OK

Line 418-420: Did this come from different parents? Could there possibly have been a family effect?

Clarified

Line 425-426: Here you could/should consider the effect of families.

Clarified

Line 433: "results in".

OK

Line 434-437: This sentence is off-topic with this paragraph – consider moving to next paragraph where you discuss proportion of genotypes at metamorphosis in Exp1 and 2. The intent of this paragraph seems to be to discuss the proportion of Pan1 genotypes at hatch in the tanks in Exp 1 and 2.

Not off-topic, but the section-division was wrong – now corrected

Lines 439-459: Why do you think the NCC genotype was favored in your experiment? Could it be because you had NCC parents (generally?). Or could it be due to the fact that you reared the fish in warmer temperature? Or water typically associated with NCC? It makes sense that the “NCC” type would be more successful in warmer temperatures, as it is a lower latitude ecotype – is this true? Would it be appropriate to discuss this point?

This is a major part of the discussion

Line 447: Could this have anything to do with the fact that you started with NCC parents?

Same as above

Line 473: There are 2 inversions adjacent to each other in LG1.

OK

Line 479: Overdominance.

OK

Line 481: spelling – necessarily.

OK

Line 483: Wahlund effect. But in this case of a controlled experiment, it does not explain excess of heterozygotes.

Through, but it is stated that this is related to nature. I think it can be as it is

Line 504-515: True. Did you control for this at all with your analysis of the parents? Could you tell which inversion type each parent had for the 2 inversions on LG1 and inversions on LG2, 7, and 12?

No, see comments at the beginning of this document

Line 522: “both cod” rather than “cod both”.

OK

Line 526: ‘were’.

OK

Line 529: In this paragraph, consider also the effect of temperature in the mesocosms. Did you measure temperature? What could it have been over the time period relative to the tanks?

Temperature now also mentioned in the discussion (in addition to in Results)

Lines 540-560: Good handling of results when you do not exactly understand why.

Figures 1 and 2: Make these consistent; consider same y-axis label, and be consistent about whether you are labeling the bars. Add “yolk sac” and metamorphosis on Figure 2 somewhere, in addition to in the caption. In Figure 2, can you state that you only used data from families 5, 6, and 7 (if that is true?).

Figures and text have been changed

Table 1: For families 2, 3, and 4 from Exp. 2 – were any of these genotyped and included in Figure 2? Did you do anything with families 2, 3, and 4? Are there any results from these families?
This is stated in the caption

Figure 3: Your legend is incomplete- what do the square, circle, and triangle represent?
OK

Supplemental Figures: Please label Supplemental Figure 1 and Supplemental Figure 2 clearly in your working draft. I did not see captions for these in the .pdf provided. If there is room in the manuscript, consider moving Supplemental Figure 1 into the main body of the paper.

Suppl. 1 is now incorporated into Figure 2. Supple 2 has been deleted.